complexity/applied mathematics

human mobility, traffic models, land use, open data, OpenStreetMap

**Author for correspondence:**
Scott A. Hale
e-mail: scott.hale@oii.ox.ac.uk

# Diagnosing the performance of human mobility models at small spatial scales using volunteered geographical information

Chico Q. Camargo[1], Jonathan Bright[1] and Scott A. Hale[1,2]

[1]Oxford Internet Institute, University of Oxford, Oxford, UK
[2]Alan Turing Institute, London, UK

CQC, 0000-0002-2947-765X; JB, 0000-0002-1248-9275; SAH, 0000-0002-6894-4951

Accurate modelling of local population movement patterns is a core, contemporary concern for urban policymakers, affecting both the short-term deployment of public transport resources and the longer-term planning of transport infrastructure. Yet, while macro-level population movement models (such as the gravity and radiation models) are well developed, micro-level alternatives are in much shorter supply, with most macro-models known to perform poorly at smaller geographical scales. In this paper, we take a first step to remedy this deficit, by leveraging two novel datasets to analyse where and why macro-level models of human mobility break down. We show how freely available data from OpenStreetMap concerning land use composition of different areas around the county of Oxfordshire in the UK can be used to diagnose mobility models and understand the types of trips they over- and underestimate when compared with empirical volumes derived from aggregated, anonymous smartphone location data. We argue for new modelling strategies that move beyond rough heuristics such as distance and population towards a detailed, granular understanding of the opportunities presented in different regions.

## 1. Introduction

Predicting human mobility is important for urban planning, traffic control, and for the general management of a city. This question has been addressed by a long tradition of mathematical models of human mobility. Most state-of-the-art population mobility

models fall under two traditions [1], namely the gravity-based models, dating back to Zipf [2] and Carey[3], and the intervening opportunities models, such as Stouffer's 1940 model [4,5], as well as the celebrated parameter-free radiation model, by Simini *et al.* [6]. In gravity models, the leading assumption is that the volume of trips going from one location to another should decrease with the distance between origin and destination. Intervening opportunity models, instead, treat trip volume as a function of the number of potential destinations between the two places.

Numerous studies have compared both classes of models [1,7–10]. In particular, a recent study using census commuting data from France, Italy, Mexico, Spain, USA, UK, and also from commuting within London and Paris, concluded that the gravity model performs better than other models at predicting traffic flows at these spatial scales, but only by a small margin [11].

Despite their many successes at predicting large-scale commuting, the gravity and radiation models are known to perform poorly in predicting traffic at spatial scales smaller than the ones described above. The modified radiation model by Yang *et al.* meant to address that issue by introducing a scaling parameter $\alpha$ representing the influence of small spatial scales in human mobility [12]. This extra parameter, as described in the original paper, would be a way to address the separation between population density and trip attraction rates that happens at small scales [12].

This problem of poor mobility predictions at small spatial scales has also been addressed by considering the variation in the accessibility of different sites [13] as well as features like the topology of urban spaces [14]. While there is no single definition of accessibility—it is often defined depending on its particular application [15–18]—it typically relates the opportunity of accessing a specific location to the cost of travel, which is essential to urban planning [18].

It makes sense to include fine-grained variables when predicting mobility at small scales, since the smaller the scale of a system the harder it becomes for its variables to average out. For example, the travel time between different neighbourhoods in a city depends not only on their distance but also on their accessibility, while at a national scale one can roughly assume all cities are equally accessible since there are major roads connecting most regions of a country and other accessibility differences might average out. Note that it remains a very rough assumption, but the point is that it becomes a safer assumption at larger scales. This behaviour is well known in the physical sciences, where larger systems also tend to be more predictable than smaller systems due to asymptotic effects such as the law of large numbers and the central limit theorem [19]. Take, for example, the diffusion of ink in a glass of water: while the motion of any small group of ink molecules might show random fluctuations, once one considers the whole ink solute, the randomness essentially disappears as the solute gradually dissolves from high-concentration areas to low-concentration areas. Thus, in chemistry, diffusion is described as a deterministic macroscopic phenomenon, despite its microscopic random nature [20]. The same applies to human mobility: at large spatial scales, the differences between different regions average out. At small spatial scales, more data are required.

Mobility models traditionally use demographic and geographical data as input, being typically limited to the population of the zones of interest and the distance between them. In the last decades, new sources of data have become available. In addition to traffic sensors that can estimate in real-time the volume of vehicles flowing through the streets, many cities are now home to a variety of sensors of traffic congestion, urban noise levels, air quality and other variables such as water and energy usage. The technology used in urban sensing also includes video surveillance powered by computer vision algorithms allowing the detection and classification of different types of vehicles as well as cyclists and pedestrians [21].

In fact, much of this sensing technology comes not in the form of dedicated devices located in strategic spots on roads and buildings, but rather integrated to mobile devices. The ever-growing number of mobile applications for urban routing, ride-sharing and sharing of geolocated information in social media, aided by the ubiquity of mobile phones, have turned these devices into 'floating sensors' [22], a valuable source of detailed geographical data. Recent studies in urban mobility have tapped into the potential of these new data sources, achieving successful traffic prediction from mobile phone and social media data alone or in combination with demographic and sensor data [1,23–25].

A third source of data is land use data. There is a vast literature on the links between land use and transport [25–27], often classifying land use types at highly aggregate levels, such as residential, commercial or industrial areas. Historically, compiling land usage datasets has been an expensive and time-consuming task [28]. The volunteered geographical information site OpenStreetMap, launched in 2004, can be seen as an alternative solution to this problem: rather than being compiled by a single person or team, OpenStreetMap is the outcome of a Wikipedia-like collaborative editing process that produces a free, open and detailed world map, including details such as the location and classification of different types of residential and

commercial structures. The accuracy and completeness of OpenStreetMap coverage has been assessed in several studies [29–37], yielding positive but cautious results, particularly about road networks. In the context of human mobility, it has been used to explain the volume of traffic incidents [24].

The diversity of new data sources for traffic prediction described above becomes a promising resource once one considers that human mobility at smaller scales is less likely to behave according to laws as simple as the ones proposed by gravity or radiation models. With that in mind, rather than using these physics-based models of traffic prediction, one can use large amounts of data to train machine learning models that hold little or no *a priori* assumptions about the system. This data-driven approach has proved quite successful in traffic prediction [24,38–40]. The problem often raised about this approach is that it does not always add explanatory power, as more complex machine learning models are known to be 'black boxes' that work quite well but are quite challenging to explain or interpret [41,42]. Additionally, these approaches have a strong historical bias, which could result in poor predictions for areas undergoing rapid development or change [43,44].

An alternative to complex, non-interpretable machine learning models are human mobility models that take into account very fine-grained details such as the topology of a city [14] or the accessibility of different sites [13,18]. In contrast to purely data-driven approaches, these models often bring with them very mechanistic assumptions about mobility, which makes it possible to evaluate such assumptions against each other. The disadvantage is that they often require knowing quite a lot about a city and use data that is not always available or easy to obtain. While there is a large increase in the availability of urban data, such data remain unevenly distributed: there is a lot of data about major cities such as London, New York, Paris and Tokyo, but much less data about smaller or poorer places [29,31,45].

This suggests that the failure of small-scale traffic prediction will be dominant in smaller cities, which often also have smaller budgets. Dedicated sensors are not always an affordable choice for local governments, and even social media data might be too scarce or expensive to obtain. In addition, ride-sharing services and navigation applications are not available in smaller cities. All these considerations make a point for models using open data, such as the OpenStreetMap described above, as an affordable way to improve traffic prediction at small spatial scales.

In this paper, we use OpenStreetMap data in combination with public demographic data to explore how different human mobility models perform at small spatial scales, comparing models against two months of mobility data in the county of Oxfordshire, UK. First, we describe the dataset and the eight classes of models used in this study. We then fit each class of models to the data and show that most models fail at predicting trips at this spatial scale.

Next, we a propose a series of standard modifications to the models, such as using travel time instead of physical distance and adding a multiplicative term to correct the underestimated urban-to-urban trip volumes. We show that these modifications do not solve the inaccuracies in the models. We argue, in agreement with the literature, that solving this problem requires more than using demographic and distance data, and show the usefulness of OpenStreetMap data as a measure of what is present in the origin and destination zones. In the end, we discuss the applicability of our method to different locations and its relation to traffic modelling at different spatial scales.

## 2. Case study: Oxfordshire, UK

Our case study examines Oxfordshire, a county in South East England, with an area of 2605 km$^2$, and a total population of approximately 680 000 inhabitants. Counties in the UK are divided into electoral wards, or simply wards, which are usually named after neighbourhoods, parishes, and other geographical marks. The names and borders of different wards are defined by the local government, and may change over time. We obtained ward-level mid-year population estimates for the 112 Oxfordshire wards, as defined in April 2016, from the UK Office of National Statistics. Shapefiles describing the border of all Oxfordshire wards were downloaded from the Digimap mapping data service [46].

For our OpenStreetMap data, we downloaded the *points of interest* from the OpenStreetMap database [47] in November 2017, which are geolocated points tagged with indications of the kind of land use found in that location. We downloaded 1 071 877 points of interest within Oxfordshire, and selected points tagged as amenity, building, land use (i.e. green spaces), leisure, office, shop and sport. All points of interest also contain 'minor' subtags, such as *pub*, *restaurant* or *cafe*, all of which might have the same major tag of *amenity*. In this paper, we chose to not use points of interest related to the transport network, such as *railway* or *street*, due to their poor coverage. We also did not consider categories with less than 100 points of interest in Oxfordshire. In total, there were 814 different types

of tags, which were used 1 106 147 times in our dataset. 174 of these types of tags had 100 or more appearances in our data (i.e. they were used to describe at least 100 different points of interest). These 174 tags account for 1 094 970 of all observed tag uses in the dataset. So in total made use of 21% of total tags but 99% of all tag uses. We used anonymized and aggregated GPS mobile phone data provided to the Oxfordshire County Council by a major smartphone operating system.

Similar data have previously been validated and successfully used in San Francisco [48] and Amsterdam [49]. The data contains estimated trip volumes for origin–destination pairs of wards in Oxford for the January and February 2017 in hourly increments. We subset the data and only use trips inferred to be made by vehicle (and not walking or cycling) and trips on weekdays made between 7.00 and 12.00.[1] We calculate the centroid of each ward and compute the geodesic distance between all centroid pairs. Finally, we obtained the travel time between the centroids of all $112 \times 112$ origin–destination pairs of wards using the Google Distance Matrix API, part of the Google Maps Platform [50].

# 3. Human mobility models

Here, we explore eight classes of human mobility models, and analyse how well they perform in predicting traffic between different wards (zones) within the county of Oxfordshire. We use four variations on the gravity model [2,3]. In its simplest form, the traffic volume $T_{ij}$ from ward $i$ to ward $j$ can be expressed as:

$$T_{ij} = A n_i^\alpha n_j^\beta f(d_{ij}), \tag{3.1}$$

where $n_i$ and $n_j$ represent the population of wards $i$ and $j$, $A$ is a normalization constant, and $f(d_{ij})$ represents a weighting cost function indicating the relation between the number of commuters $T_{ij}$ and the commuting cost $d_{ij}$, which is typically the distance or travel time between wards $i$ and $j$. This dependency is typically modelled $f(d_{ij}) = d_{ij}^{-\gamma}$ or $f(d_{ij}) = e^{-\gamma d_{ij}}$. Finally, $\alpha$, $\beta$ and $\gamma$ are tunable parameters. The gravity model is often fit by constraining the total traffic going in ($\sum_i T_{ij}$) and out ($\sum_j T_{ij}$) of every ward to have the same values as in the ground truth data, in what is called the doubly constrained gravity model [51]. A common, simpler version of the model, more similar to the Newtonian law of gravity, is found by setting the exponents of $n_i$ and $n_j$ to $\alpha = 1$ and $\beta = 1$ [52]. In this paper, we use four versions of the gravity model, given by the four permutations of the power function and the exponential function for $f(d_{ij})$ and whether $\alpha$ and $\beta$ are set to 1 or whether they are fit to the data. While fixing $\alpha$ and $\beta$ to values other than 1 or only fixing one of the two parameters is possible, such variations are rare in the literature and not an avenue we pursue in this paper.

In addition to the gravity models discussed above, our analysis also includes the parameter-free radiation model by Simini *et al.* [6]. In this model, shown in equation (3.2), the total traffic going from $i$ to $j$ depends not only on the population in both wards, i.e. $n_i$ and $n_j$, but also on the number of opportunities between $i$ and $j$, indicated by $s_{ij}$, and measured as the total population within a circle of radius $r = d_{ij}$, where $d_{ij}$ is the distance between wards $i$ and $j$. This $s_{ij}$ term, sometimes described as the number of intervening opportunities [5], can be seen as a proxy for the number of alternative places where a person from ward $i$ could go with less cost than it would be to travel to ward $j$. Consequently, large $s_{ij}$ would lead to lower $T_{ij}$. The name for this term is no coincidence, as the radiation model was partially inspired by the intervening opportunities model [6]. The radiation model is defined as

$$T_{ij} = T_i \frac{n_i n_j}{(n_i + s_{ij})(n_i + n_j + s_{ij})}. \tag{3.2}$$

In the equation above, $T_i = \sum_j T_{ij}$. We also include Yang *et al.*'s one-parameter radiation model [12]

$$T_{ij} = T_i \frac{[(n_i + n_j + s_{ij})^\alpha - (n_i + s_{ij})^\alpha](n_i^\alpha + 1)}{[(n_i + s_{ij})^\alpha + 1][(n_i + n_j + s_{ij})^\alpha + 1]}. \tag{3.3}$$

In this model, $\alpha$ measures the influence of small spatial scales in human mobility, by changing the coupling between population density and trip attraction rates at small distances [12]. We include two versions of the modified radiation model: one with the parameter $\alpha$ fit to the data with no constraints, and another with its value defined according to the average size of Oxfordshire wards using the formula in [12] of $\alpha = (l/36[\text{km}])^{1.33}$, where $l$ stands for the average size of Oxfordshire wards.

---

[1]We also experimented using the whole day and including weekend trips, but the overall results were qualitatively similar.

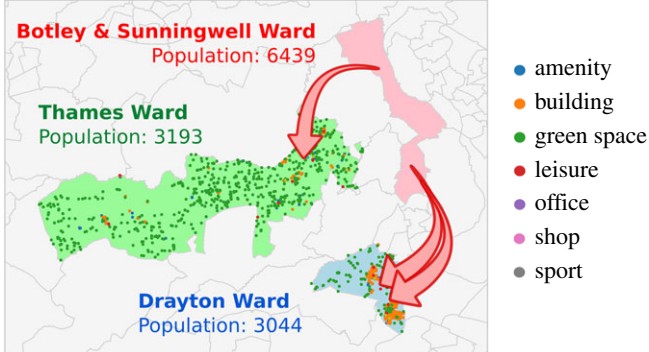

**Figure 1.** Illustration of the difference in trip volumes between wards in Oxfordshire. In the figure, both destinations (Thames Ward and Drayton Ward) have similar resident and working populations, and both are at approximately 11 km from Botley & Sunningwell Ward. However, there are about twice as many trips from Botley to Drayton as there are from Botley to Thames. This might be explained by the difference in the composition of both wards as shown by their points of interest on OpenStreetMap: Drayton Ward is markedly more urban compared to Thames Ward.

Finally, we also include the intervening opportunities model, as formulated by Schneider [5], one of the original inspirations of the radiation model [6]. The intervening opportunities model is defined as

$$T_{ij} = e^{-\gamma s_{ij}} - e^{-\gamma(s_{ij}+n_j)}. \tag{3.4}$$

Models with parameters to be fit to the data are fit using the normalized root mean squared error between $\log T_{ij}$ from the model predictions and from the ground-truth mobility data, using methods from the Python scipy package [53]. Parameter values and log NRMSE scores are shown in appendix E, and the performance of all eight models with the Oxfordshire traffic data is presented in figure 2.

# 4. Mobility at small spatial scales

Figure 1 illustrates a typical pair of trips within the Oxfordshire dataset. The figure shows three wards in Oxfordshire: Botley & Sunningwell Ward, Thames Ward and Drayton Ward. The latter two have similar resident and working populations, and both are at approximately 11 km from Botley & Sunningwell Ward (distance measured from their centroids). The number of opportunities between the wards, expressed as $s_{ij}$ in the models in §3, is the only variable in the original models that varies between the Botley–Drayton trip and the Botley–Thames origin–destination pairs: it is 60% higher for the Botley–Drayton trip, which would suggest this origin–destination pair would have a lower volume of traffic than the Botley–Thames pair. The data, however, show the opposite trend: the traffic volume for the Botley–Drayton trip is approximately twice the volume for the Botley–Thames trip.

Out of many factors that could possibly explain the higher trip volume for the Botley–Drayton trip, one simple explanation jumps out when plotting the OpenStreetMap points of interest for the Drayton and Thames Wards. While Thames Ward is filled with points of interest classified as *green space*, with no significant density as in other OSM categories, Drayton Ward shows a large number of points marked as *green space*, but also a large number of points classified as *building*, indicating urban locations. These urban centres correspond to the villages of Drayton and Milton, which together account for most of the population of Drayton Ward. On the other hand, Thames Ward is composed of a handful of smaller villages and hamlets, which altogether make a much less urban environment despite a similar total population.

# 5. Model performance at small spatial scales

When used to predict mobility data for Oxfordshire, all classes of mobility models presented in §3 underestimate high-volume trips. As the 2% origin–destination pairs with the highest trip volume correspond to 14% of all traffic volume, these trips cannot simply be discarded.

Figure 2a shows the predicted trip volume $T_{ij}^{\mathrm{model}}$ for every volume versus the trip volume $T_{ij}^{\mathrm{data}}$, both plotted in logarithmic scale. Larger volume trips are consistently underestimated, having their $y$-values below the diagonal line marking $T_{ij}^{\mathrm{model}}/T_{ij}^{\mathrm{data}} = 1$. In the figure, grey dots indicate the 2% trips with the highest volume, while the coloured dots indicate the other 98%. Considering that the models presented in

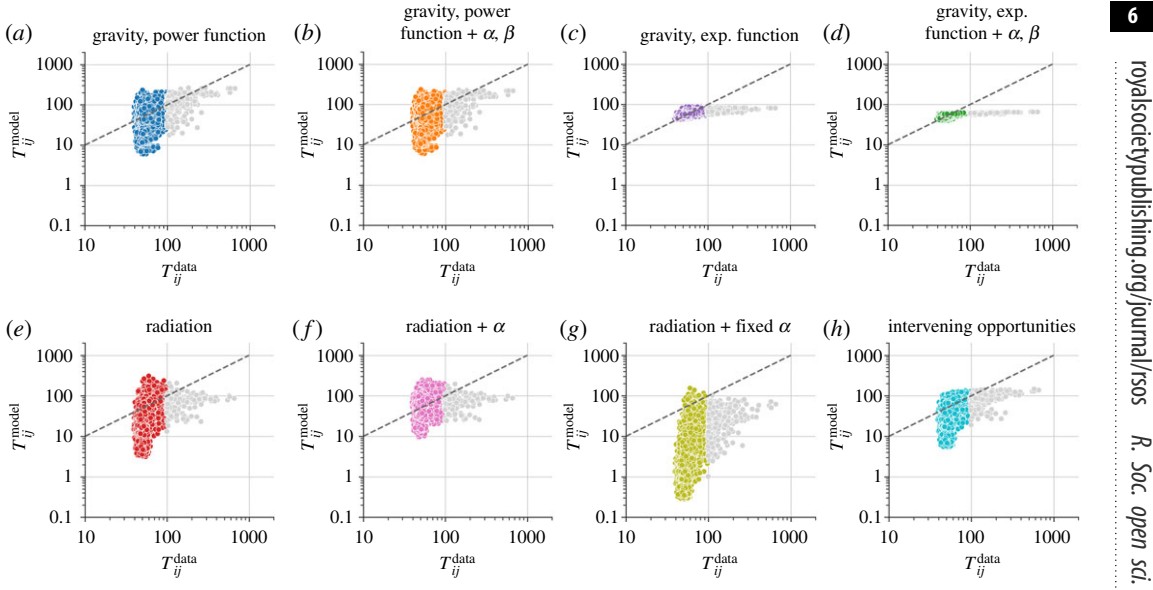

**Figure 2.** Current mobility models do not perform well at small spatial scales. (*a*–*h*) The predicted trip volume $T_{ij}^{\text{model}}$ versus the measured trip volume $T_{ij}^{\text{data}}$ according to smartphone mobility data, both plotted in logarithmic scale, for all models described in §3. All models show a poor fit to the data, especially for high-volume trips. Grey dots indicate the 2% trips with the highest volume, which correspond to 14% of all traffic volume, and dashed lines indicate the $y = x$ identity line.

§3 only take into account the population of different wards and the distance between their centroids, it is not surprising that they ignore the details that make Thames Ward different from Drayton Ward, which lie not in their aggregate demographics, but in their composition in terms of points of interest. Still, one might suggest these differences could be addressed by incorporating measures of accessibility such as travel time [15,54]. The reason for this is straightforward: for short trips, travel distance and duration do not scale linearly (see appendix A), and thus distance is not always a good proxy for accessibility. One could also try a more straightforward modification to the models, by adding a multiplicative factor to modify the predicted volume for shorter trips. Drawing inspiration from the $\text{e}^{-s_{ij}}$ term in Stouffer's implementation of the intervening opportunities model, and considering that the number of intervening opportunities $s_{ij}$ grows with distance and travel time, we propose multiplying $T_{ij}$ by a term depending on $s_{ij}$ only, in such a way that large $s_{ij}$ are not penalized, but low $s_{ij}$ lead to an increase in $T_{ij}$. With this extra term, which includes free parameters to be fit by the data, we write a modified expression for the trip volume $T_{ij}$

$$T_{ij}^{\text{mod}} = T_{ij} \times (1 + A\text{e}^{-s_{ij}/b}). \tag{5.1}$$

These two possible model modifications, i.e. using travel time in place of distance and having the extra dependency on $s_{ij}$, when considered separately or jointly, make a total of three additional models in addition to the original unmodified $T_{ij}$. These three possible modifications, when combined with the eight types of mobility models presented above, give a total of 32 possible mobility models: the eight original models, plus 24 possible modifications (three per model).

We assess the performance of different models using two traditional goodness-of-fit measures for human mobility models. The first is the Common Part of Commuters (CPC) [55,56], based on the Sørensen index [57]

$$\text{CPC}(T^{\text{model}}, T^{\text{data}}) = \frac{2 \sum_{i,j} \min(T_{ij}^{\text{model}}, T_{ij}^{\text{data}})}{\sum_{i,j} T_{ij}^{\text{model}} + \sum_{i,j} T_{ij}^{\text{data}}}. \tag{5.2}$$

When the total number of commuters $N$ is preserved, the expression for the CPC can be written as a function that decreases with the absolute difference between $T^{\text{model}}$ and $T^{\text{data}}$

$$\text{CPC}(T^{\text{model}}, T^{\text{data}}) = 1 - \frac{1}{2N} \sum_{i,j} |T_{ij}^{\text{model}} - T_{ij}^{\text{data}}|. \tag{5.3}$$

**Figure 3.** Simple modifications do not improve model performance. Goodness of fit heatmaps for Common Part of Commuters (CPC) and Common Fraction of Commuters (CFC) for all combinations of the eight models against the four modifications, including the non-modified models. Columns labelled as `Original`, T, S and T+S, respectively, represent the original models, followed by models fit using travel time rather than travel distance, models including the $s_{ij}$ modification, and models including both modifications. The maximum and minimum values for 95% confidence intervals generated from 10 000 bootstrap samples differed by less than $10^{-4}$ and hence are not shown.

We also define the Common Fraction of Commuters (CFC), a measure analogous to CPC, but for the ratio between $T_{ij}^{\mathrm{model}}$ and $T_{ij}^{\mathrm{data}}$, in a way that higher values of $T_{ij}$ do not skew the sum, and every $(i, j)$ pair counts equally, regardless of the traffic volume on that origin–destination pair

$$\mathrm{CFC}(T^{\mathrm{model}}, T^{\mathrm{data}}) = \frac{1}{N} \sum_{i,j} \min\left(\frac{T_{ij}^{\mathrm{model}}}{T_{ij}^{\mathrm{data}}}, \frac{T_{ij}^{\mathrm{data}}}{T_{ij}^{\mathrm{model}}}\right). \tag{5.4}$$

The expression for the CFC can also be written as a function that decreases with the absolute difference between $\log T_{ij}^{\mathrm{model}}$ and $\log T_{ij}^{\mathrm{data}}$. We provide the full derivation in the appendix C, and the final expression here

$$\mathrm{CFC}(T^{\mathrm{model}}, T^{\mathrm{data}}) = \frac{1}{N} \sum_{i,j} \exp\left(-\frac{1}{2}\left|\log T_{ij}^{\mathrm{data}} - \log T_{ij}^{\mathrm{model}}\right|\right). \tag{5.5}$$

The comparison of how well the 32 models described above predict trips in Oxfordshire can be seen in figure 3. Figure 3a,b show values for CPC and CFC, respectively, for all combinations of the eight models against the three modifications, plus the unmodified models. Each row represents one of the model classes presented in §3, and columns labelled with `Original`, T, S and T+S, respectively, represent the original models, followed by models fit using travel time rather than distance, models including the $s_{ij}$ modification, and models including both modifications.

The first result to be noticed from this figure is that both CPC and CFC result in very similar heat maps. Even though the two measures might differ in their individual values, they follow an approximately linear relation where CFC ≈ 1.018 × CPC − 0.104, as shown in appendix B. Still, according to both measures, the best fit is obtained by the gravity models with exponents $\alpha$ and $\beta$ fit to the data.

Besides the difference between both goodness of fit metrics, one can also compare different columns within the same heatmap. Overall, changes in CPC and CFC are minimal, usually less than 1%, except for cases when the original value was already low. There is almost no difference between the non-modified models (`Original`) and the models modified with travel time (T), which suggests that swapping travel distance by travel time does not have any noticeable effect on trip prediction in our setting. The $s_{ij}$ modification alone (S) produces a large improvement in both CPC and CFC for the one-parameter radiation models—particularly with a fixed $\alpha$ value—, while having a small effect on all other models (less than 1% CPC and CFC). Using both modifications to the models, (T+S) offers no further improvements. The one-parameter radiation models with fixed $\alpha$ show an improvement over the original models but that improvement is accounted for by the modified $s_{ij}$ term alone. The intervening opportunities model shows a decrease in performance with both modifications, while for the majority

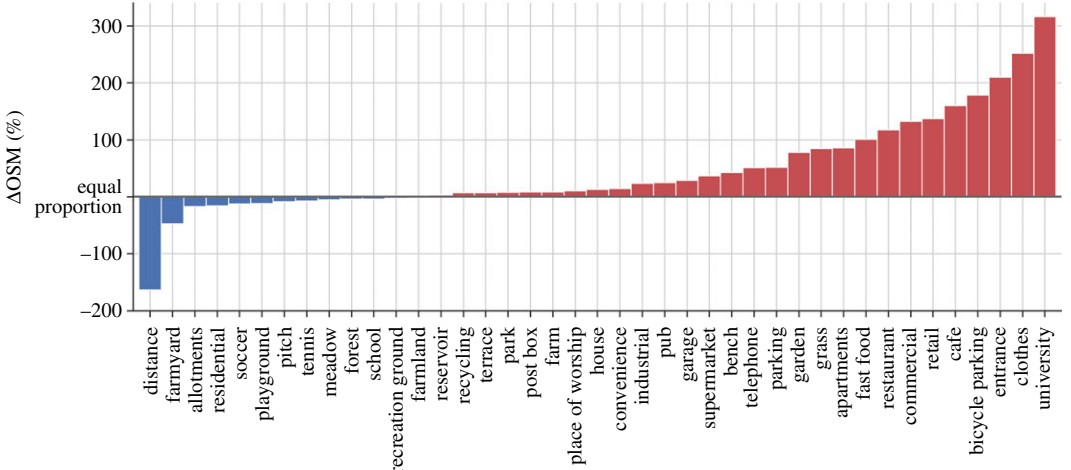

**Figure 4.** The average high-volume trip is a short-distance trip between urban areas. In this plot, the length of every bar shows the relative difference between a trip attribute (travel distance, or the density of an OSM tag at the destination ward) when averaged over the 2% trips with the highest volume, versus the same attribute when averaged over the other 98% of trips. Trip attributes which are more frequent for the 2% high-volume trips are shown in red, whereas trip attributes more frequent for the other 98% of trips are shown in blue. For example, the density of points of interest tagged as *fast food* or *restaurant* is approximately 100% higher for high-volume trips, while the average travel distance is 150% higher for the other group. All attributes shown in the plot vary at least 1% between both groups.

of models, the difference is within $\pm 1\%$. We also estimated 95% confidence intervals by bootstrap sampling 10 000 samples of size 10 000 each with replacement. As the width of all confidence intervals was under $10^{-4}$, we did not show them in figure 3.

Overall, figure 3 shows that replacing the distance between the centroids of each ward by the travel time between them or trying to artificially increase the predicted trip volume for short trips by modifying the functional form of human mobility models does not solve the fact that current mobility models do not perform well at this spatial scale, as presented in figure 2.

# 6. Using OSM data to diagnose the problem

The first step towards identifying where current mobility models fail is to observe their prediction accuracy is considerably different for the trips of the highest volume, which lie on the right side of every panel in figure 2. In the eight panels, the grey dots indicate the 2% trips with the highest volume, while the coloured dots indicate the other 98%. Having split our dataset into two groups, we can then investigate whether the trips in both groups differ in any other attributes, such as their average travel distance, number of intervening opportunities ($s_{ij}$), or the density of any point of interest present in OpenStreetMap. This is a way of addressing the difference illustrated by the trips to Drayton Ward and Thames Ward, as presented in figure 1, where two origin–destination pairs of comparable distance and population actually led towards of very different composition in terms of OpenStreetMap points of interest.

Figure 4 compares the composition of destination wards between the top 2% high-volume trips, shown in red, versus the remaining 98% of trips, shown in blue.[2] The figure shows the average distance travelled between wards, as well as the average density of a series of OSM tags in the destination ward, for all OSM categories whose density varies more than 1% between both groups of trips. In this plot, the length of a bar shows the relative difference between the average value of a trip attribute for all high-volume trips from the average value for all other trips. For example, the number of points of interest tagged as *clothes* is 250% higher for the destinations of high-volume trips compared to the destinations of all other trips. Similarly, the average distance of high-volume trips is approximately 150% lower than that of all other trips. In short, the figure shows that high-volume trips are often short-distance trips between urban centres.

Figure 5 shows how OpenStreetMap data can be used to reveal where human mobility models fail. For the remainder of this section, we focus on the best-performing model, the gravity model with a power

---

[2]Tags corresponding to origin wards result in lower values, but show a similar trend.

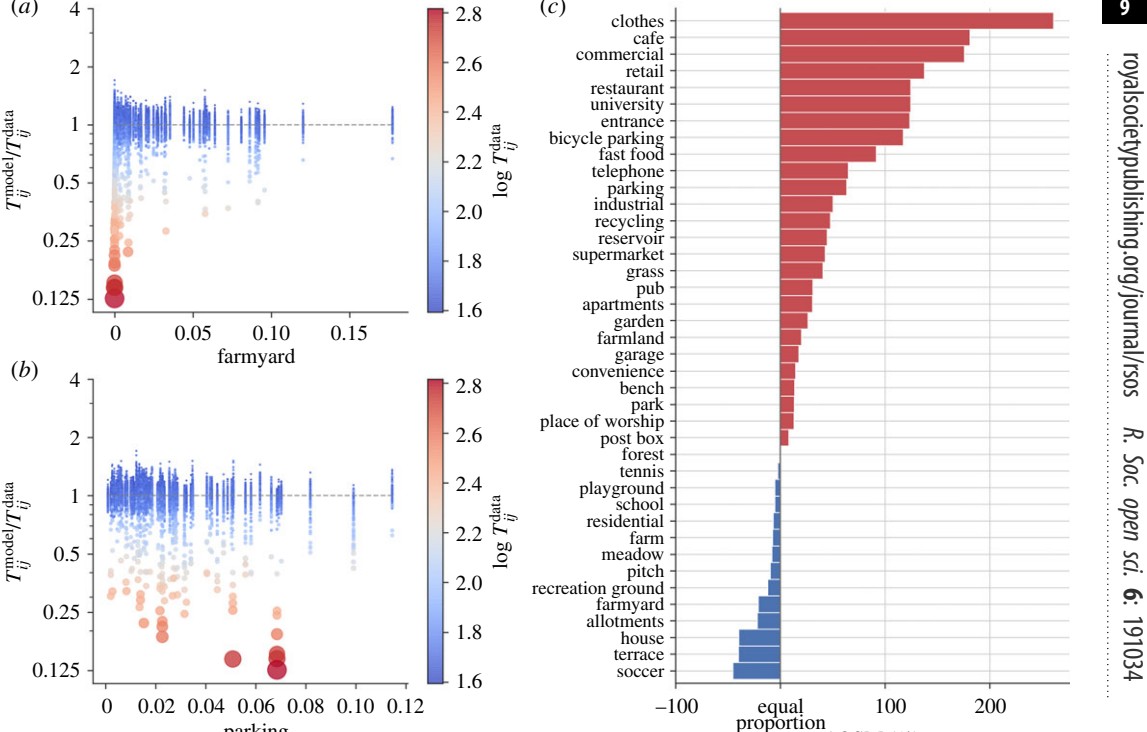

**Figure 5.** High-volume urban trips are underestimated by human mobility models. (*a,b*) Scatter plots for the ratio $T_{ij}^{\text{model}}/T_{ij}^{\text{data}}$ against the density of the OSM tags *farmyard* and *parking* in the destination ward, respectively. The $T_{ij}^{\text{model}}/T_{ij}^{\text{data}}$ ratio indicates the overestimation error for the model class with the best performance, namely the gravity model with power functions on the travel cost and parameters fit to the data. Dot size and colour indicate the magnitude of the real trip volume ($T_{ij}^{\text{data}}$), with small blue dots for low volumes and large red dots for large volumes. The bar plot in (*c*) shows the difference between the trip attributes of the 2% trips with the lowest overestimation ratio when compared with the other 98% of trips. For example, the density of POIs tagged as *cafe* is nearly 200% higher for the underestimated trips, while the density of points POIs tagged as *soccer* is approximately 50% higher for the non-overestimated trips.

function and $\alpha$ and $\beta$ fit to the data (without any modifications), but the analysis could be conducted with any mobility model. Figure 5*a,b* shows scatter plots for the ratio $T_{ij}^{\text{model}}/T_{ij}^{\text{data}}$, which indicates the error made by the model. A ratio greater than one indicates overestimation and lower than one indicates underestimation. In both plots, dot size and colour indicate the magnitude of the real trip volume $T_{ij}^{\text{data}}$, with small blue dots indicating low trip volumes and large red dots indicating large trip volumes.

On figure 5*a*, the overestimation ratio is plotted against the density of the OpenStreetMap minor tag *farmyard* in the destination ward, i.e. the number of occurrences of points of interest with that tag, divided by ward population. Most trips are placed around $T_{ij}^{\text{model}}/T_{ij}^{\text{data}} = 1$, meaning little or no overestimation and cover the whole range of values in the *x*-axis, corresponding towards with a varying density of points of interest tagged as *farmyard*. By comparison, the high-volume trips, marked by the red dots, are all placed at the lowest densities of *farmyard* points of interest, and the volumes of these trips are consistently underestimated.

The same underestimation of high-volume trips is present in figure 5*b*, which shows the density of points of interest tagged as *parking* along the *x*-axis. In this case, the trend for high-volume trips is the opposite of the trend present in figure 5*a*: while most trips are distributed over the whole range of *parking* density, high-volume trips are biased towards the higher end of the range. The comparison between figure 5*a,b* suggests that the destination wards of high-volume trips are more likely to have parking lots and less likely to have farmyards.

The analysis present in figure 5*a,b* can be repeated for any other OpenStreetMap tag. Figure 5*c* shows the result of producing the same scatter plots for the overestimation as presented in the other panels, but for all the OpenStreetMap tags shown in figure 4. This bar plot is produced after first splitting trip predictions two groups: one group has the 2% trips with the lowest overestimation ratio, corresponding to trips with $T_{ij}^{\text{model}}/T_{ij}^{\text{data}} < 0.686$, and the other group has the remaining 98% of trips

with an overestimation ratio above this value.[3] In this bar plot, as in figure 4, the length of a bar shows the relative difference between the average value of a trip attribute or number of points of interest between the two groups, in this case, the 2% most underestimated trips on the left and the rest of the dataset on the right. For example, the density of points of interest tagged as *cafe* is nearly 200% higher for the highly underestimated trips, while the density of points of interest tagged as *soccer* is approximately 50% higher for the non-overestimated trips. As a whole, the bar plot indicates that the model underestimates urban trips, characterized by a higher density of points of interest tagged as clothes shops, cafes and general retail.

# 7. Discussion

In this paper, we presented eight classes of human mobility models, including the gravity, radiation and intervening opportunity models. After calibrating all models using two months of aggregate and anonymous smartphone mobility data for the county of Oxfordshire, UK, we showed that most models give poor predictions of traffic volume at this spatial scale. We also showed that typical modifications applied to mobility models, such as exchanging travel distance for travel time or adding extra parameters to correct for poor performance at small spatial scales, do not generally result in any significant improvement in the model fits, which we measured using the CPC and the CFC. While the CPC and CFC scores were not particularly poor, our analysis showed that the 2% of trips with the highest volume were consistently underestimated by even the best-performing model.

Of course, one possibility raised by our results is that OSM data could be used not only to diagnose problems with mobility models but potentially also provide means of improving them. The possibility of enhancing traffic models with more data is in fact explored in a number of recent works, which have used new data sources such as road configurations [58], city block shapes [14], travel cost and number of opportunities [18], among many variables which provide a more fine-grained description of how regions of a city differ from each other. We believe these approaches are important, as they repeatedly show that describing human mobility at small spatial scales in urban environments requires further differentiating between regions which might otherwise be similar at an aggregate level. The approach described here could be used to help test and diagnose possible modifications such as those mentioned above.

OSM data specifically could provide a number of potential further enhancements in this area. For example, a count of OSM points of interest (or other features such as buildings) could substitute $n_j$ in the gravity model (equation (3.1) above), replacing the idea of people 'gravitating' towards other people with the idea of them gravitating towards features which might be of interest to them. Relatedly, these feature counts could also be used to estimate the number of intervening opportunities between two points of interest ($s_{ij}$ in the models described in equations (3.2) and (3.3) above). Indeed, one could even imagine making use of granular types of points of interest for measuring different types of population flow: for example, office blocks and industrial estates might indicate commuting, while shopping and leisure activities might be used to estimate weekend travel. These avenues would be promising subjects for future work.

Our method using OpenStreetMap data provides a more thorough diagnosis of where mobility models fail than aggregate error measures such as CPC and CFC. In Oxfordshire, our analysis revealed that the trips of highest volume have a specific profile in terms of their density of points of interest. The 2% trips with the highest volume trips have a high density of points labelled as university facilities, clothes and other forms of retail, cafes, restaurants, and other urban landmarks, with densities sometimes over 200% higher than all other trips. High-volume trips are also typically over much shorter distances.

We also showed that those high-volume trips are severely underestimated even by the best-performing mobility models, sometimes being predicted at only approximately 12.5% of the actual traffic volume. Taking the mobility model with the best performance for further analysis, we split trips into the 2% most underestimated and the remaining 98%, and show that the origin–destination pairs for which the model performs worst are indeed the most urban ones, characterized by the same OpenStreetMap tags as the set of trips of highest volume.

In this paper, we show how to use OpenStreetMap data to diagnose where and how mobility models fail, but it may be possible to also use OpenStreetMap data to improve mobility models. Population (or

---

[3]The top 2% of overestimated trips is similar to the top 2% of trips by volume. The overlap between these two sets is 94% over all origin–destination pairs.

working population) is often used to calculate the attractiveness of a region (i.e. $n_i$ and $n_j$) as well as for calculating the number of intervening opportunities ($s_{ij}$). Our work suggests that OpenStreetMap points of interest and their distributions, as a fine-grained indications of land use, could be promising to directly incorporate into mobility models. This is an idea, however, that we leave for future work.

Unlike data sources that might only be available at a specific time or location or that might incur the expense of deploying large numbers of urban sensors, the OpenStreetMap dataset used in this paper is freely available worldwide. As long as there is enough coverage, OpenStreetMap data can be a powerful and easily applicable tool when combined with other demographic and geographical data to diagnose the performance of multiple human mobility models. This is in stark contrast with more expensive approaches requiring traffic sensors or opaque methods, which might not be viable options for local governments, for reasons of time, skills, or budgetary constraints. Assuming some ground truth data is available, the diagnosis method we describe here does not require any expensive computation, sensor deployment or proprietary data. As explained above, this method is flexible and easily applicable to multiple human mobility models as long as there is enough OpenStreetMap coverage in the region. Who contributes to OpenStreetMap and the biases of the data are important to consider. While more urban or populous locations are likely to have more detailed coverage [45,59,60], volunteered geographical information is still much more widespread and available than fixed-sensor, ride-sharing and other similar data.

This work also adds to the discussion of how to model human mobility at different spatial scales. While the dataset in this paper describes traffic within the perimeter of a region of roughly 2605 km$^2$, mobility models have been tested and compared across different scales [13,61], including their many successful applications describing cross-country migration [6,23,56,62]. At a national scale, many of the modifications discussed here should not be necessary, since travel time scales with travel distance for larger scales (see appendix A), but also since large, city-sized regions are likely to contain wider distributions of OpenStreetMap points of interest, rather than the very uneven distributions observed for different Oxfordshire wards.

Given the importance of understanding human mobility at small spatial scales for urban planning, it is vital to have tools that can flexibly incorporate diverse sources of land use and accessibility data into model diagnosis and traffic prediction. Here we have shown how OpenStreetMap can be one such tool, providing useful insights to the study of how people move in urban landscapes.

Data accessibility. Data are available from Zenodo at https://zenodo.org/record/3383443.

Authors' contributions. All authors conceived and designed the study and collected the data. C.Q.C. implemented the models, carried out the analysis and wrote the first draft. S.A.H. and J.B. secured the funding and S.A.H. coordinated the project. All authors edited the manuscript and gave final approval for publication.

Competing interests. The authors declare no competing financial interests.

Funding. This project was supported by funding from Innovate UK under grant no. 52277-393176, the NERC under grant no. NE/N00728X/1, and the Lloyd's Register Foundation and The Alan Turing Institute under the EPSRC grant no. EP/N510129/1.

# Appendix A. Scaling of measures of travel distance

Figure 6 shows how different metrics for travel distance scale with each other. Figure 6a shows how trip duration in minutes and the distance in kilometres between wards scales in an almost linear fashion, with a Pearson coefficient of $r = 0.94$ ($p < 10^{-16}$) differing only for trips shorter than 10 km or less than 15 min driving. Two fits are also shown on the figure: a linear fit showing trip duration can be approximated by $\delta t_{ij} = 0.95 \times d_{ij} + 10.87$ and a nonlinear fit estimating trip duration as $\delta t_{ij} = 4.99 \times d_{ij}^{0.61}$. For both fits, $\delta t_{ij}$ is the trip duration measured in minutes, and $d_{ij}$ is the distance in kilometres between wards $i$ and $j$.

Figure 6b shows trip duration in minutes on the $x$-axis and the number of intervening opportunities between wards $i$ and $j$, given by the variable $s_{ij}$, on the $y$-axis. The plot shows a clear positive correlation between both variables ($r = 0.86$, $p < 10^{-16}$). In figure 6a,b, smaller lighter dots indicate trips with a lower trip volume (i.e. lower $T_{ij}$), while larger darker dots indicate higher trip volumes.

# Appendix B. Scaling of CPC with CFC

Figure 7 shows the scaling between the CPC and the CFC for the model variations shown in figure 3. The variables follow an approximately linear relation, with CFC $\approx 1.018 \times$ CPC $- 0.104$.

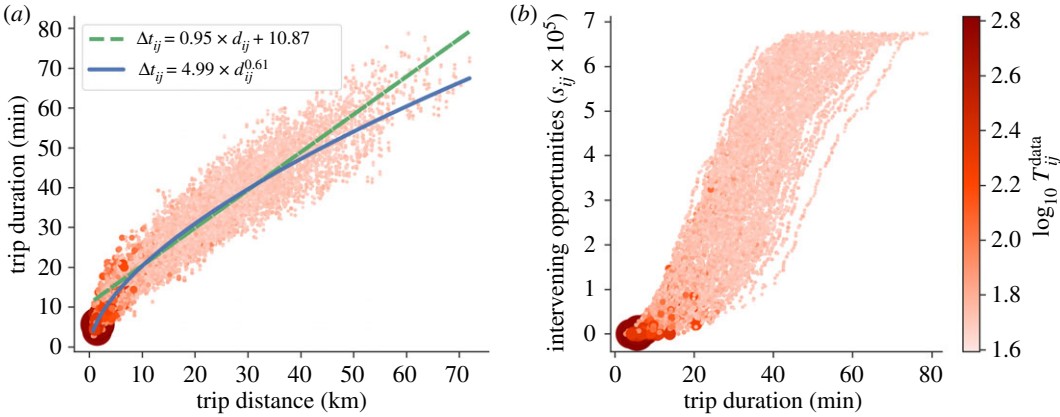

**Figure 6.** Scaling between different metrics for travel distance. (a) How trip duration measured in minutes and travel distance measured in kilometres scale in an almost linear fashion. (b) Trip duration on the x-axis, and intervening opportunities ($s_{ij}$) on the y-axis. The plot shows a clear positive correlation between both variables ($r = 0.86$, $p < 10^{-16}$). In both panels, smaller lighter dots indicate trips with a lower trip volume, while larger darker dots indicate higher trip volumes.

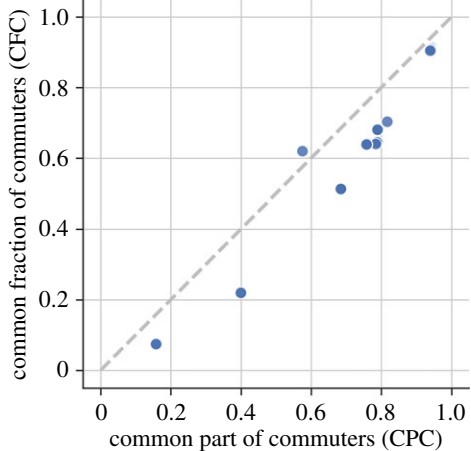

**Figure 7.** Scaling between the Common Part of Commuters (CPC) and the Common Fraction of Commuters (CFC) for the model variations shown in figure 3. The variables follow an approximately linear relation, with CFC $\approx 1.018 \times$ CPC $- 0.104$.

## Appendix C. Derivation of how CFC is the L1 norm on log $T_{ij}$

When defining the CFC, we state that equation (5.4) can also be written as a function of $|\log T_{ij}^{\text{data}} - \log T_{ij}^{\text{model}}|$, as shown in equation (5.5). This can be seen below

$$\frac{1}{N}\sum_{ij}\min\left(\frac{T_0}{T_1}, \frac{T_1}{T_0}\right) = \frac{1}{N}\sum_{ij}\min\left(e^{\ln T_0 - \ln T_1}, e^{\ln T_1 - \ln T_0}\right)$$

$$= \frac{1}{N}\sum_{ij}\exp\left[\min\left(\ln T_0 - \ln T_1, \ln T_1 - \ln T_0\right)\right]$$

$$= \frac{1}{N}\sum_{ij}\exp\left(-\frac{1}{2}|\ln T_0 - \ln T_1|\right).$$

The last line uses the fact that $\min(x, y) = (x + y - |x - y|)/2$. Note the last line is a sum of absolute-value differences between $\ln T_0$ and $\ln T_1$, making CFC close to a L1 norm on the log $T_{ij}$.

## Appendix D. The threshold for high traffic is robust

Figure 8 shows a series of bar plots similar to figure 4, but for different high-volume cut-offs, ranging from 2% to 8%. It shows no meaningful variation in the properties of the high-volume trips and the

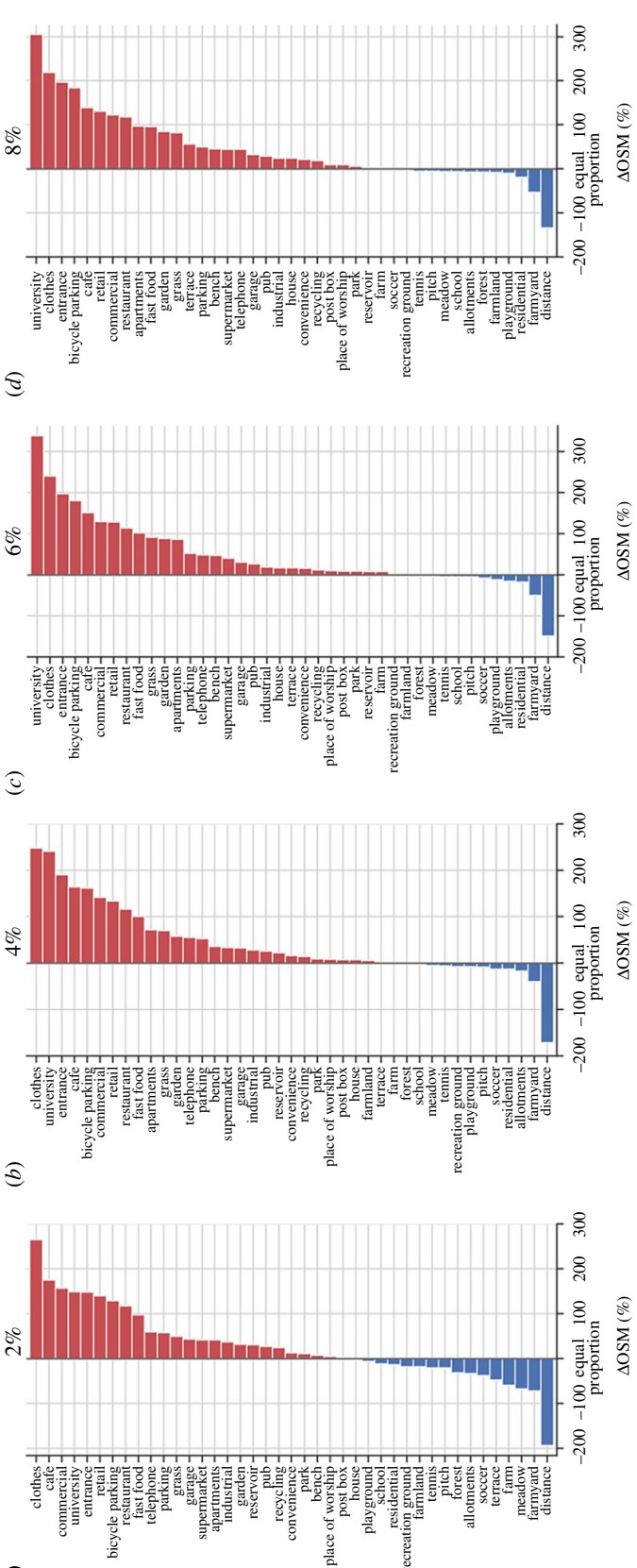

**Figure 8.** The high-volume split is robust to the cut-off threshold. (*a*–*d*) Bar plots similar to figure 4, but for different cut-offs, ranging from 2% to 8%. They show no meaningful variation in the properties of the high-volume trips and the properties of all other trips in terms of OpenStreetMap points of interest. In all cases, high-volume trips are on average over shorter distances and to more urban areas.

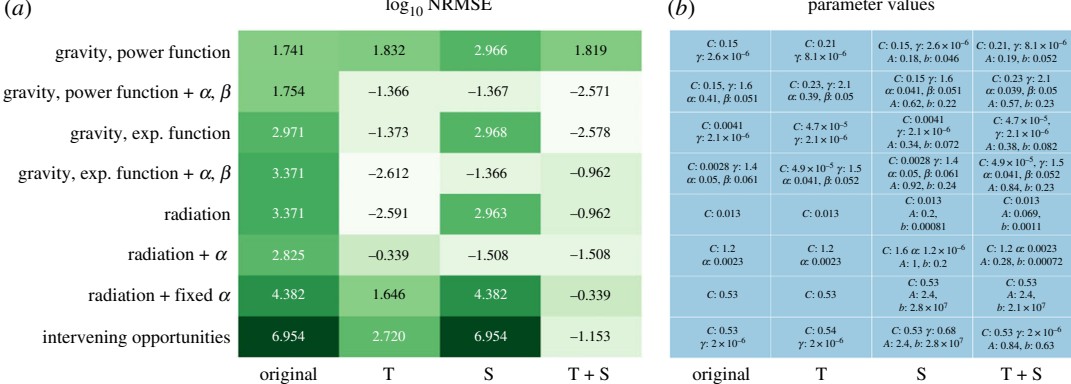

**Figure 9.** NRMSE and parameter values for all models. (*a*) The heatmap shows the $\log_{10}$ NRMSE for all combinations of the eight models against the four modifications, including the non-modified models. (*b*) The parameters of every model, ordered in the same way, with the *C* indicating a normalization parameter, present in all models. Columns labelled as `Original`, `T`, `S` and `T+S`, respectively, represent the original models, followed by models fit using travel time, models including the $s_{ij}$ modification, and models with both modifications.

properties of all other trips in terms of OpenStreetMap points of interest. In all cases, high-volume trips are on average shorter and go to more urban places than do other trips.

# Appendix E. Parameter values and NRMSE scores

The heatmap in figure 9*a* shows how $\log_{10}$ NRMSE varies over all combinations of the eight models against the four modifications. Figure 9*b* shows the final parameter values obtained when fitting each of the models.

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
