## [Reviewer comments · Royal Society Open Science]

Review History

RSOS-191034.R0 (Original submission)

Review form: Reviewer 1

Is the manuscript scientifically sound in its present form?

No

Are the interpretations and conclusions justified by the results?

No

Is the language acceptable?

Yes

Do you have any ethical concerns with this paper?

No

Have you any concerns about statistical analyses in this paper?

Yes

Recommendation?

Major revision is needed (please make suggestions in comments)

Comments to the Author(s)

This paper uses GPS data from a major smartphone operating system combined with geographical information from a freely available app, OpenStreetMap, to analyse the behaviour of various population movement models. The goal of the paper is to identify where and why these models fail to describe human mobility at small scales, which appears to be a known phenomenon. The authors consider variations of two well-known models, the gravity and radiation models, obtaining eight models in total. After showing that all the models perform poorly when predicting traffic volumes at small scales, the authors propose some modifications to these models (leading to 24 new models), which in turn do not result in any significant improvement on the previous results. This leads to the conclusion that macroscopic models must include information about what sort of amenities are present in the origin and destination zones, for which OpenStreetMap is useful.

The paper succeeds in identifying certain characteristics of high-volume trips, which (I believe) should become an indicator for when the above-mentioned models would fail. However, I believe the paper would be greatly improved with more detailed explanations of the methodologies, which would make it more suitable for a wide audience.

I suggest the authors consider the following comments:

1) General comments:

- Are the gravity and radiation models the state of the art for traffic volume? Are there any other models which are not mentioned, and if so, why were these particular ones chosen?
- A comparison between the gravity, radiation and intervening opportunities models would be useful to highlight the main differences and similarities between the models, and give some intuition on why one would expect any of them to perform better/worse than the others.
- The authors define s_{ij} as the number of interest points between i and j that might make people from i go somewhere else rather than to j . It seems to me that the paper's main conclusion is that instead, one should use a different parameter which would count the number of interest points in j that would make people from i go to j rather than somewhere else (i.e, go to Drayton Ward rather than Thames Ward). Is this something that could be easily modified in the proposed models, and subsequently tested with the methodology presented in this paper?
- Similarly, the authors comment on other modifications proposed in the literature (second paragraph of conclusions); can these models be tested with the methodology presented in this paper?

2) Comments on the models, methodology and presentation:

- I am not clear on what the authors mean by "... alpha and beta are (...) free to vary" for the gravity model. Does this mean that the parameters vary within the simulation? Or are different for different wards? Are they fit to the data (and if so, what do the authors use for the fit?). The alternative to this seems to be fixing $\alpha = \beta = 1$. Is there any reason why they can't be any other constant (or even different constants)?
 - Same as above, but for the radiation model and "with the parameter alpha unconstrained": what does unconstrained mean?
 - The variable T_i in equations (3.2) and (3.3) is not defined anywhere in the paper.
 - Should the estimated values for alpha, beta and gamma be presented (maybe in an appendix)?
- In general, the statistical analysis performed to obtain the results should be explained. The authors should be clear on what parameters are being fitted and state their estimated values. Furthermore, the authors mention that they fit the parameters using mean squared error, would it be useful to have a comparison of the MSE for each case as a measure of how good or bad the

fit is? An explanation of how the confidence intervals are calculated (definition of bootstrap samples) would be beneficial as well.

- Figure 2 and its explanation:

o The figure is not fully explained until Section 6. There, the authors refer to a dashed red line which separates the 2% trips with highest volume from the rest – these lines don't exist, and I assume the authors mean grey dots?

o In section 5 it is mentioned that Figure 2a shows $T^{\text{model}}/T^{\text{data}}$ vs T^{data} ; I believe this should be T^{model} vs T^{data} (as stated in the caption of Fig 2)? Otherwise it would not make sense to plot both in a logarithmic scale.

- Figure 3 and its explanation:

o I don't understand what is meant by "and all other trips" in line 20, page 8. All other trips compared to what?

o The sentence "resulting in values $<10^{-4}$ for all estimates" is used a few times. What are these values?

o The authors say that the modification (S) makes results typically worse, but I would say that this is not the case: it does make gravity + exponential + alpha/beta and intervening opportunities worse, but it makes the radiation models better, which is not noted in the text.

o Page 8, lines 40-45, the authors mention that adding (T) decreases CPC but not CFC, but this is not the case: whenever CPC is decreased, CFC is too (and this is not for the six models mentioned). Furthermore, the authors mention "the differences between (S) and (T)" but only (T) is mentioned after this.

o The changes in CPC and CFC are minimal (usually less than 1%, unless the original model was really bad), should this be commented upon?

Other minor comments:

- Page 7, line 47, should common part of commuters be capitalised (like Common Fraction of Commuters in line 57)?

- Page 8, line 37, were should be where.

- Page 11, line 41 "Taking the the..."

- Page 11, line 50 "this is in stark contrast more expensive..." (with missing?)

Review form: Reviewer 2

Is the manuscript scientifically sound in its present form?

Yes

Are the interpretations and conclusions justified by the results?

Yes

Is the language acceptable?

Yes

Do you have any ethical concerns with this paper?

No

Have you any concerns about statistical analyses in this paper?

No

Recommendation?

Accept with minor revision (please list in comments)

Comments to the Author(s)

Overall, this is a very interesting paper which focuses on an important research question. The authors provide a careful discussion of a range of human mobility models, together with their strengths and weaknesses. The analysis is well performed.

I have a few minor comments that I would like the authors to address:

- in section 2, they state that categories with less than a hundred points of interest in Oxfordshire are not considered; how many categories were left after removing those? I think a little bit more information on this section would be beneficial for the reader
- the explanation of the parameter-free radiation model in section 3 is not very clear
- my main comment is regarding figures 4 and 5; as they currently are, they are rather difficult to interpret and take quite a long time before they become clear and intuitive. I don't have a specific suggestion on how to change them, and the authors are welcome to leave the figures as they are if they wish, but it is worth mentioning that they are not obvious to interpret.
- the authors should at least discuss (if not provide an analysis of) some ways of including OpenStreetMap data in order to improve predictions/models of human mobility. I think it's really interesting to understand how existing models fail, and using OpenStreetMap to do so is an excellent idea, but it would be interesting to know how this data can be included in order to enhance existing models.

In general, this is a good manuscript which could be suitable for publication with some small changes.

Decision letter (RSOS-191034.R0)

05-Aug-2019

Dear Dr Quevedo Camargo,

The editors assigned to your paper ("Diagnosing the performance of human mobility models at small spatial scales using volunteered geographic information") have now received comments from reviewers. We would like you to revise your paper in accordance with the referee and Associate Editor suggestions which can be found below (not including confidential reports to the Editor). Please note this decision does not guarantee eventual acceptance.

Please submit a copy of your revised paper before 28-Aug-2019. Please note that the revision deadline will expire at 00.00am on this date. If we do not hear from you within this time then it will be assumed that the paper has been withdrawn. In exceptional circumstances, extensions may be possible if agreed with the Editorial Office in advance. We do not allow multiple rounds of revision so we urge you to make every effort to fully address all of the comments at this stage. If deemed necessary by the Editors, your manuscript will be sent back to one or more of the original reviewers for assessment. If the original reviewers are not available, we may invite new reviewers.

To revise your manuscript, log into <http://mc.manuscriptcentral.com/rsos> and enter your

Author Centre, where you will find your manuscript title listed under "Manuscripts with Decisions." Under "Actions," click on "Create a Revision." Your manuscript number has been appended to denote a revision. Revise your manuscript and upload a new version through your Author Centre.

- Data accessibility

If you wish to submit your supporting data or code to Dryad (<http://datadryad.org/>), or modify your current submission to dryad, please use the following link:
<http://datadryad.org/submit?journalID=RSOS&manu=RSOS-191034>

- Competing interests

- Authors' contributions

AB carried out the molecular lab work, participated in data analysis, carried out sequence alignments, participated in the design of the study and drafted the manuscript; CD carried out the statistical analyses; EF collected field data; GH conceived of the study, designed the study,

coordinated the study and helped draft the manuscript. All authors gave final approval for publication.

- Acknowledgements

- Funding statement

on behalf of Dr Jose Carrillo (Associate Editor) and Mark Chaplain (Subject Editor)
openscience@royalsociety.org

Reviewers' Comments to Author:

Reviewer: 1

Comments to the Author(s)

This paper uses GPS data from a major smartphone operating system combined with geographical information from a freely available app, OpenStreetMap, to analyse the behaviour of various population movement models. The goal of the paper is to identify where and why these models fail to describe human mobility at small scales, which appears to be a known phenomenon. The authors consider variations of two well-known models, the gravity and radiation models, obtaining eight models in total. After showing that all the models perform poorly when predicting traffic volumes at small scales, the authors propose some modifications to these models (leading to 24 new models), which in turn do not result in any significant improvement on the previous results. This leads to the conclusion that macroscopic models must include information about what sort of amenities are present in the origin and destination zones, for which OpenStreetMap is useful.

The paper succeeds in identifying certain characteristics of high-volume trips, which (I believe) should become an indicator for when the above-mentioned models would fail. However, I believe the paper would be greatly improved with more detailed explanations of the methodologies, which would make it more suitable for a wide audience.

I suggest the authors consider the following comments:

1) General comments:

- Are the gravity and radiation models the state of the art for traffic volume? Are there any other models which are not mentioned, and if so, why were these particular ones chosen?
- A comparison between the gravity, radiation and intervening opportunities models would be useful to highlight the main differences and similarities between the models, and give some intuition on why one would expect any of them to perform better/worse than the others.

- The authors define s_{ij} as the number of interest points between i and j that might make people from i go somewhere else rather than to j . It seems to me that the paper's main conclusion is that instead, one should use a different parameter which would count the number of interest points in j that would make people from i go to j rather than somewhere else (i.e, go to Drayton Ward rather than Thames Ward). Is this something that could be easily modified in the proposed models, and subsequently tested with the methodology presented in this paper?
- Similarly, the authors comment on other modifications proposed in the literature (second paragraph of conclusions); can these models be tested with the methodology presented in this paper?

2) Comments on the models, methodology and presentation:

- I am not clear on what the authors mean by "... alpha and beta are (...) free to vary" for the gravity model. Does this mean that the parameters vary within the simulation? Or are different for different wards? Are they fit to the data (and if so, what do the authors use for the fit?). The alternative to this seems to be fixing $\alpha = \beta = 1$. Is there any reason why they can't be any other constant (or even different constants)?
- Same as above, but for the radiation model and "with the parameter alpha unconstrained": what does unconstrained mean?
- The variable T_i in equations (3.2) and (3.3) is not defined anywhere in the paper.
- Should the estimated values for alpha, beta and gamma be presented (maybe in an appendix)? In general, the statistical analysis performed to obtain the results should be explained. The authors should be clear on what parameters are being fitted and state their estimated values. Furthermore, the authors mention that they fit the parameters using mean squared error, would it be useful to have a comparison of the MSE for each case as a measure of how good or bad the fit is? An explanation of how the confidence intervals are calculated (definition of bootstrap samples) would be beneficial as well.
- Figure 2 and its explanation:
 - o The figure is not fully explained until Section 6. There, the authors refer to a dashed red line which separates the 2% trips with highest volume from the rest - these lines don't exist, and I assume the authors mean grey dots?
 - o In section 5 it is mentioned that Figure 2a shows $T^{\text{model}}/T^{\text{data}}$ vs T^{data} ; I believe this should be T^{model} vs T^{data} (as stated in the caption of Fig 2)? Otherwise it would not make sense to plot both in a logarithmic scale.
- Figure 3 and its explanation:
 - o I don't understand what is meant by "and all other trips" in line 20, page 8. All other trips compared to what?
 - o The sentence "resulting in values $<10^{-4}$ for all estimates" is used a few times. What are these values?
 - o The authors say that the modification (S) makes results typically worse, but I would say that this is not the case: it does make gravity + exponential + alpha/beta and intervening opportunities worse, but it makes the radiation models better, which is not noted in the text.
 - o Page 8, lines 40-45, the authors mention that adding (T) decreases CPC but not CFC, but this is not the case: whenever CPC is decreased, CFC is too (and this is not for the six models mentioned). Furthermore, the authors mention "the differences between (S) and (T)" but only (T) is mentioned after this.
 - o The changes in CPC and CFC are minimal (usually less than 1%, unless the original model was really bad), should this be commented upon?

Other minor comments:

- Page 7, line 47, should common part of commuters be capitalised (like Common Fraction of Commuters in line 57)?
- Page 8, line 37, were should be where.
- Page 11, line 41 "Taking the the..."

- Page 11, line 50 "this is in stark contrast more expensive..." (with missing?)

Reviewer: 2

Comments to the Author(s)

Overall, this is a very interesting paper which focuses on an important research question. The authors provide a careful discussion of a range of human mobility models, together with their strengths and weaknesses. The analysis is well performed.

I have a few minor comments that I would like the authors to address:

- in section 2, they state that categories with less than a hundred points of interest in Oxfordshire are not considered; how many categories were left after removing those? I think a little bit more information on this section would be beneficial for the reader

- the explanation of the parameter-free radiation model in section 3 is not very clear

- my main comment is regarding figures 4 and 5; as they currently are, they are rather difficult to interpret and take quite a long time before they become clear and intuitive. I don't have a specific suggestion on how to change them, and the authors are welcome to leave the figures as they are if they wish, but it is worth mentioning that they are not obvious to interpret.

- the authors should at least discuss (if not provide an analysis of) some ways of including OpenStreetMap data in order to improve predictions/models of human mobility. I think it's really interesting to understand how existing models fail, and using OpenStreetMap to do so is an excellent idea, but it would be interesting to know how this data can be included in order to enhance existing models.

In general, this is a good manuscript which could be suitable for publication with some small changes.

Author's Response to Decision Letter for (RSOS-191034.R0)

See Appendix A.

RSOS-191034.R1 (Revision)

Review form: Reviewer 1

Is the manuscript scientifically sound in its present form?

Yes

Are the interpretations and conclusions justified by the results?

Yes

Is the language acceptable?

Yes

Do you have any ethical concerns with this paper?

No

Have you any concerns about statistical analyses in this paper?

No

Recommendation?

Accept as is

Comments to the Author(s)

The authors have addressed all my comments and I am happy for the paper to be published in its current state.

Review form: Reviewer 2

Is the manuscript scientifically sound in its present form?

Yes

Are the interpretations and conclusions justified by the results?

Yes

Is the language acceptable?

Yes

Do you have any ethical concerns with this paper?

No

Have you any concerns about statistical analyses in this paper?

No

Recommendation?

Accept as is

Comments to the Author(s)

The authors have addressed all comments raised in my previous review, I think the manuscript is now suitable for publication.

Decision letter (RSOS-191034.R1)

02-Oct-2019

Dear Dr Quevedo Camargo,

I am pleased to inform you that your manuscript entitled "Diagnosing the performance of human

mobility models at small spatial scales using volunteered geographic information" is now accepted for publication in Royal Society Open Science.

Best regards,
Lianne Parkhouse
Royal Society Open Science
openscience@royalsociety.org

on behalf of Dr Jose Carrillo (Associate Editor) and Mark Chaplain (Subject Editor)
openscience@royalsociety.org

Reviewer comments to Author:

Reviewer: 1
Comments to the Author(s):

The authors have addressed all my comments and I am happy for the paper to be published in its current state

Reviewer: 2
Comments to the Author(s):

The authors have addressed all comments raised in my previous review, I think the manuscript is now suitable for publication.

Appendix A

Response to the reviewers

We thank the reviewers for their thoughtful assessment of our work. In the following we list the points raised and how we have addressed each point.

Reviewer 1

This paper uses GPS data from a major smartphone operating system combined with geographical information from a freely available app, OpenStreetMap, to analyse the behaviour of various population movement models. The goal of the paper is to identify where and why these models fail to describe human mobility at small scales, which appears to be a known phenomenon. The authors consider variations of two well-known models, the gravity and radiation models, obtaining eight models in total. After showing that all the models perform poorly when predicting traffic volumes at small scales, the authors propose some modifications to these models (leading to 24 new models), which in turn do not result in any significant improvement on the previous results. This leads to the conclusion that macroscopic models must include information about what sort of amenities are present in the origin and destination zones, for which OpenStreetMap is useful.

The paper succeeds in identifying certain characteristics of high-volume trips, which (I believe) should become an indicator for when the above-mentioned models would fail. However, I believe the paper would be greatly improved with more detailed explanations of the methodologies, which would make it more suitable for a wide audience.

I suggest the authors consider the following comments:

1) General Comments:

Reviewer Point P 1.1 — Are the gravity and radiation models the state of the art for traffic volume? Are there any other models which are not mentioned, and if so, why were these particular ones chosen?

Reply: In the absence of sensor or other mobility data, the gravity and radiation models along with various modifications to them are state-of-the-art. We have now added to the introduction a more detailed explanation of state-of-the-art human mobility models. It now says “Most state-of-the-art population mobility models fall under two traditions (Barbosa et al., 2018), namely the gravity-based models (...) and the intervening opportunities models [which includes the radiation model].”

Reviewer Point P 1.2 — A comparison between the gravity, radiation and intervening opportunities models would be useful to highlight the main differences and similarities between the models, and give some intuition on why one would expect any of them to perform better/worse than the others.

Reply: We have now added to the introduction a description of the differences between these classes of models and of how they perform at different scales.

Reviewer Point P 1.3 — The authors define s_{ij} as the number of interest points between i and j that might make people from i go somewhere else rather than to j . It seems to me that the paper’s main conclusion is that instead, one should use a different parameter which would count the number of interest points in j that would make people from i go to j rather than somewhere else (i.e, go to Drayton Ward rather than Thames Ward). Is this something that could be easily modified in the proposed models, and subsequently tested with the methodology presented in this paper?

Reply: Thank you for this suggestion. s_{ij} is defined as the number of intervening opportunities between i and j , which is usually measured based on the population (or work working population) of locations between i and j . Similarly all the models use the populations of i and j as a measure of their respective attractivenesses. Indeed, we do think that making some use of the OpenStreetMap POI data in the way described (i.e., offering an actual count of amenities rather than using the total population as a proxy) should offer a potential to improve on the existing models we propose. We also think that this is feasible. However, as there are many different potential ways of operationalising T_{ij} under this idea: we prefer to keep the focus of the paper on the diagnosis of existing problems with mobility models and present this as an avenue for further research. We have amended the discussion to reflect this line of thinking, and also included a bit of a wider discussion of potential uses of OSM data following suggestion 2.4 from reviewer 2 (page 12).

Reviewer Point P 1.4 — Similarly, the authors comment on other modifications proposed in the literature (second paragraph of conclusions); can these models be tested with the methodology presented in this paper?

Reply: The modifications referenced there such as city block shapes (Brelsford et al. 2018) require specific datasets which, unfortunately, we do not have access to in the specific area of our study (Oxfordshire). Therefore it is not possible for us to test these modifications. Currently these modifications are not widely used or accepted (which which was another reason we chose not to test them or include them in our work), but we hope others may use our approach in the future to specifically investigate these and other modifications. We have added language to this effect within that paragraph indicating that a valuable contribution of our approach is to help test and diagnose possible modifications such as those mentioned.

2) Comments on the models, methodology and presentation:

Reviewer Point P 1.5 — I am not clear on what the authors mean by “... alpha and beta are (...) free to vary” for the gravity model. Does this mean that the parameters vary within the simulation? Or are different for different wards? Are they fit to the data (and if so, what do the authors use for the fit?). The alternative to this seems to be fixing $\alpha = \beta = 1$. Is there any reason why they can’t be any other constant (or even different constants)?

Reply: By “unconstrained” and “free to vary,” we meant that we fit these parameters to the data, and we have edited the text to say “fit to the data” in place of these phrases. We have also moved the following sentence from *Model performance at small spatial scales* to *Human mobility models* and increased its prominence to make our fitting procedure clear.

Models with parameters to be fit to the data are fit using the mean squared error between $\log T_{ij}$ from the model predictions and from the ground-truth mobility data, using methods from the Python scipy package.

We also added the following sentence:

While fixing α and β to values other than 1 or only fixing one of the two parameters is possible, such variations are rare in the literature and not an avenue we pursue in this paper.

Reviewer Point P 1.6 — Same as above, but for the radiation model and “with the parameter alpha unconstrained”: what does unconstrained mean?

Reply: We have similarly updated the text to read “fit to the data” and thank the reviewer for helping us more clearly communicate our models.

Reviewer Point P 1.7 — The variable T_i in equations (3.2) and (3.3) is not defined anywhere in the paper.

Reply: We now define it right after equation (3.2).

Reviewer Point P 1.8 — Should the estimated values for alpha, beta and gamma be presented (maybe in an appendix)? In general, the statistical analysis performed to obtain the results should be explained. The authors should be clear on what parameters are being fitted and state their estimated values. Furthermore, the authors mention that they fit the parameters using mean squared error, would it be useful to have a comparison of the MSE for each case as a measure of how good or bad the fit is? An explanation of how the confidence intervals are calculated (definition of bootstrap samples) would be beneficial as well.

Reply: We now provide estimated values for all parameters in an appendix and also report the $\log_{10}(\text{NRMSE})$ in the same appendix. We did not report these values originally as we felt the specific values for Oxfordshire would be less interesting for readers, but on reflection agree that reporting them in an appendix could be helpful for reprehensibility and transparency. We have also modified the language describing our bootstrap sampling approach as well (further details below).

Reviewer Point P 1.9 — Figure 2 and its explanation: The figure is not fully explained until Section 6. There, the authors refer to a dashed red line which separates the 2% trips with highest volume from the rest – these lines don’t exist, and I assume the authors mean grey dots?

Reply: We do mean grey dots indeed. We have rewritten that part, and it now says “In the eight panels, the grey dots indicate the 2% trips with the highest volume, while the coloured dots indicate the other 98%.” We have also added this description when the figure is first described in Section 5.

Reviewer Point P 1.10 — [Figure 2 and its explanation] In section 5 it is mentioned that Figure 2a shows T^{model}/T^{data} vs T^{data} ; I believe this should be T^{model} vs T^{data} (as stated in the caption of Fig 2)? Otherwise it would not make sense to plot both in a logarithmic scale.

Reply: That is correct. We have fixed it, and it now says “Figure 2a shows the predicted trip volume T_{ij}^{model} for every volume versus the trip volume T_{ij}^{data} , both plotted in logarithmic scale.”

Reviewer Point P 1.11 — Figure 3 and its explanation: I don’t understand what is meant by “and all other trips” in line 20, page 8. All other trips compared to what?

Reply: That passage was indeed unclear. We have changed our figure descriptions to “Even though the two measures might differ in their individual values, they follow an approximately linear relation where $CFC \approx 1.018 \times CPC - 0.104$, as shown in Appendix B”, for Figure 3, and “Figure 4 compares the composition of destination wards between the top 2% high-volume trips, shown in red, versus the remaining 98% of trips, shown in blue’, for Figure 4.’

Reviewer Point P 1.12 — [Figure 3 and its explanation] The sentence “resulting in values $< 10^{-4}$ for all estimates” is used a few times. What are these values?

Reply: We thank the reviewer for pointing out the confusion here. We intended to communicate that the 95% confidence intervals are small and the maximum and minimum values of these confidence intervals differ by less than 10^{-4} . We have updated the caption to more clearly communicate this. We have also updated the text in the paper on this point and to describe the bootstrap sampling process more clearly.

Reviewer Point P 1.13 — [Figure 3 and its explanation] The authors say that the modification (S) makes results typically worse, but I would say that this is not the case: it does make gravity + exponential + alpha/beta and intervening opportunities worse, but it makes the radiation models better, which is not noted in the text.

Reply: We agree, and have rewritten that sentence. It now reads “The s_{ij} modification alone (S) produces a large improvement in both CPC and CFC for the one-parameter radiation models—particularly with a fixed α value—, while having a small effect on all other models ($< 1\%$ CPC and CFC).”

Reviewer Point P 1.14 — [Figure 3 and its explanation] Page 8, lines 40-45, the authors mention that adding (T) decreases CPC but not CFC, but this is not the case: whenever CPC is decreased, CFC is too (and this is not for the six models mentioned). Furthermore, the authors mention “the differences between (S) and (T)” but only (T) is mentioned after this.

Reply: We have now restructured that paragraph, describing both (S) and (T).

Reviewer Point P 1.15 — [Figure 3 and its explanation] The changes in CPC and CFC are minimal (usually less than 1%, unless the original model was really bad), should this be commented upon?

Reply: We agree and now specifically comment on this in the text.

Minor points

Reviewer Point P 1.16 — Page 7, line 47, should common part of commuters be capitalised (like Common Fraction of Commuters in line 57)?

Reply: Fixed.

Reviewer Point P 1.17 — Page 8, line 37, were should be where.

Reply: Fixed.

Reviewer Point P 1.18 — Page 11, line 41 “Taking the the...”.

Reply: Fixed.

Reviewer Point P 1.19 — Page 11, line 50 “this is in stark contrast more expensive...” (with missing?)

Reply: Fixed: added “with”.

Reviewer 2

Overall, this is a very interesting paper which focuses on an important research question. The authors provide a careful discussion of a range of human mobility models, together with their strengths and weaknesses. The analysis is well performed.

I have a few minor comments that I would like the authors to address:

Reviewer Point P 2.1 — In section 2, they state that categories with less than a hundred points of interest in Oxfordshire are not considered; how many categories were left after removing those? I think a little bit more information on this section would be beneficial for the reader.

Reply: We agree we needed more detail here. In total there were 814 different types of tags, which were used 1,106,147 times in our dataset. 174 of these types of tags had 100 or more appearances in our data (i.e., they were used to describe at least 100 different points of interest). These 174 tags account for 1,094,970 of all observed tag uses in the dataset. So in total we code $174/814 = 21\%$ of total tags and $1,094,970/1,106,147 = 99\%$ of all tag uses. We have amended the text to make this clear in Section 2 on page 4.

Reviewer Point P 2.2 — The explanation of the parameter-free radiation model in section 3 is not very clear.

Reply: We have now described it in more detail.

Reviewer Point P 2.3 — My main comment is regarding figures 4 and 5; as they currently are, they are rather difficult to interpret and take quite a long time before they become clear and intuitive. I don't have a specific suggestion on how to change them, and the authors are welcome to leave the figures as they are if they wish, but it is worth mentioning that they are not obvious to interpret.

Reply: Thank you for this comment. We have revised the figure descriptions in an attempt to make them easier to interpret. We unfortunately could not find an alternative figure design that more clearly communicated the information.

Reviewer Point P 2.4 — The authors should at least discuss (if not provide an analysis of) some ways of including OpenStreetMap data in order to improve predictions/models of human mobility. I think it's really interesting to understand how existing models fail, and using OpenStreetMap to

do so is an excellent idea, but it would be interesting to know how this data can be included in order to enhance existing models.

Reply: Thank you. We more explicitly address this in the *Discussions* now. (Please also see our response to P1.3 above.) Incorporating OSM data into mobility models is an ongoing interest of ours, but one which we do not feel can be adequately addressed within the scope of this paper. We have, however, enhanced the discussion section to specifically mention this possibility and provide the readers with some potential suggestions.

Closing comments

We would like to close by once again thanking the reviewers for so deeply and thoughtfully engaging with our work. In addition to the specific points above, we have carefully reviewed the manuscript and clarified our presentation of the data, methods, and results wherever possible. Overall, we feel the reviewers' comments and the resulting modifications have greatly strengthened the manuscript and are very grateful to the reviewers for reviewing our revised manuscript.